# Analysis of Circular Thinking in Consumer Purchase Intention to Buy Sustainable Waste-To-Value (WTV) Foods

**Shahjahan Ali** [1] , **Shahnaj Akter** [2] **and Csaba Fogarassy** [3,*]

1 Doctoral School of Economic and Regional Sciences, Hungarian University of Agriculture and Life Sciences, 2100 Gödöllő, Hungary; ali.shahjahan@hallgato.uni-szie.hu
2 Business School, Beijing Normal University, Beijing 100875, China; shahnaj.eco@gmail.com
3 Institute of Sustainable Development and Farming, Hungarian University of Agriculture and Life Sciences, 2100 Gödöllő, Hungary
* Correspondence: fogarassy.csaba@uni-mate.hu

**Abstract:** One of the new fronts in food research is related to waste reuse and the impact of by-products on food nutrition intensity. These foods are Waste-to-Value (WTV) products that are suitable for demonstrating the processes of the circular economy (CE), in which another excess material is converted into a new food, generating higher nutritional properties. The manifestation of customer reaction is very strong when buying these products. Consumer findings can strongly support or hinder the development of circular systems through our purchasing decisions. In this way, it is essential to evaluate consumer WTV foods to learn about related consumer habits. Consumers can support or hinder the circular economy with their purchasing intentions. This analysis's primary objective is to evaluate what different factors can be applied to consumers' perception in purchasing sustainable WTV foods towards CE. In this study, a well-constructed questionnaire was prepared. Five hundred and forty-four (544) people participated in the survey, of which, 499 samples were analyzed. The primary research question was, "Would the consumer buy a sustainable Waste-to-Value (WTV) food product that affects the environment when it is produced? That is, it does not come from a circular system?" The other question is, how do the origin of products, information on production/nutritional value, consumer education, and certain socio-demographic characteristics affect the value of waste value for sustainable food consumption? According to the research results, in the case of the surveyed consumers, the younger age group (18–35 years old) shows a greater preference for buying sustainable products. It is also a surprising and new result that gender characteristics in this age group do not influence consumption patterns. Women and men showed the same preferences. Our second hypothesis is that education positively affects consumer intentions for sustainable WTV foods and especially organic products. The questionnaire did not confirm this.

**Keywords:** waste-to-value food; circular economy; consumer intention; sustainable consumption; nutritional value; consumer gender issues; consumer education issues

## 1. Introduction

Sustainable Production (SP) and Sustainable Consumption (SC) are necessary conditions for the feasible turn of events [1]. They are characterized by the United Nations (UN) and improving them [2]. SP is essential for sustainable consumption that fulfills consumer requirements as diminishing negative sustainability influence [3,4]. Later, the connections between sustainable consumption and the circular economy (CE) are more concentrated [5]. Beyond a wide range of definitions of the circular economy, the fundamental objective of the methodology is to keep away from and limit product and asset utilization through different material circles [6]. As per the circular economy concept, the valuation of the product and raw material ought to be kept up to the extent that this would be possible, for instance, limiting waste or utilizing it to make value-added items. Thus, the CE approach would change production cycles to fulfil consumer demands in new

and more sustainable ways [7]. Like other economic activities, production and consumption substantially manipulate the local ecosystem through resource usage besides waste production. Food production puts significant pressure on the environment, especially through the use of water, energy, pesticides and fertilizers. Their manageability is considered to be well manageable in circular systems [8]. To encourage a CE method in food production, one viewpoint that calls for consideration is the improvement and recycling of materials in every case they return into the production network. In numerous agricultural creations/products, deposits can trigger genuine manageability issues considering the high amounts by a product delivered in a restricted period (perishable foods), and the general substance in natural point, which is wasted [9,10]. The valorization of waste and the food supply chain results in an unexpected sustainability issue in agriculture, as it yields numerous economic, environmental, and social advantages [11]. One of the new limits of research on agricultural food in this area concerns the reuse of waste and the effects of by-products on food nutrition intensity [12]. These food items, like virgin olive oil, are Waste-to-Value (WTV) products [13]. To feature CE methodology that reuses waste otherwise considered excess materials [14], obtained throughout the other processes of food manufacturing, into newly added food in conjunction with higher properties of nutrition [15]. The production of olive oil is excellent due to this unique circumstance. Middle East nations are the principal producers of olive oil. People consume a higher quantity of olive in these countries [16,17]. The processing of olive oil reproduces a large amount of food waste. However, among these waste products, olive leaves, still wealthy in bioactive mixes that could recuperate from getting high worth-added food items [18,19], thereby assisting with spreading the CE in the area [20,21]. Olive tree leaves are a wellspring of cell reinforcement phenolic intensification, notable for their potential medical advantages [22]. The healing of phenolic removes from olive leaves has broad research in progress. The acquired concentrates can be utilized to advance new food items, such as practical fixings or characteristic cell reinforcement added substances [23]. For almost all companies engaged in marketing on both offline and online channels, the price has always been at the lead of promotions. However, all relevant stakeholders are interested in the effect of these changes on purchase behavior. In India, a product's price based on a bid was 2.8 times higher than the actual cost of the product. According to this report, Indian consumers use actual price as a quality metric [24]. In Italy, the knowledgeable buyer of biofortified goods is adamant about buying them and is willing to pay a premium. Consumers who are aware and well-informed make up a small percentage of the industry. The purchasing and consumption of biofortified foods are influenced by the various knowledge assets available to consumers. According to the report, campaigns to communicate health-related properties may play a major role in shaping market dynamics [25]. Knowledge spillovers seem to influence the dynamic mix of work relocation and wage factors that drive innovation. Marshallian spillovers negatively influence the green economy because they confirm the prevalence of the displacement effect [26]. However, once these WTV food items have been created, their last market take-up relies upon consumer purchase decisions. Especially in the agricultural food manufacturing area, customer reaction is undeniable in improving higher nutrition. The findings of the consumer can support or hamper the CE on account of their last buying choices. In this manner, assessing consumers' value of WTV food is essential to evaluate such a narrative diet's inevitable market achievement [27]. This investigation's primary objective is to assess the applicability of various components that favor the consumer's commitment to the CE by buying sustainable WTV food products. For this study, a structured questionnaire was prepared. Five hundred forty-four (544) people participated in the survey. Due to a missing value, the study used a 499-strong sample. The survey was conducted by google-docs in an online environment, with direct interviews and e-mail. A binary logistic regression model was assessed in arrangement to evaluate the possible drivers of statements of consumers. This paper aims to determine the overall significance of the inevitable elements of influencing consumer buying for sustainable WTV (waste-to-value) food in this specific situation. Some diverse goals of

consumer purchase were investigated. The first question is, "Would the consumer buy a sustainable Waste-to-Value (WTV) food product that negatively affects the environment when it is produced? That is, it does not come from a circular system?" This subsequent inquiry was believed to be suitable to uncover an express ecological inspiration driving the consumer's purchase intention. The writing on purchasers' acknowledgement for food got from side-effects are restricted why this explorable region is very new, and there are not many items, effectively created, that can be tried [13,15,17]. There is excellent literature on novel nourishments' acknowledgement by customers and buyers' inclinations for even more biologically efficient items. This examination intensely expands upon the discoveries of these fields of exploration. The first is consumer purchase intention of sustainable WTV food products characterized as purchasers who have a proper inclination to new food evasion [28]. Numerous examinations have indicated that purchase intention influences both the quality and variety of diets charred around [29,30].

An increasingly common approach to the development of sustainable business is the Circular Economy (CE). A CE seeks to achieve a healthy environment and economy through numerous product and material loops by minimizing resource use [31]. However, there are many different definitions. A general purpose of what is known as a circular business model is only slowly emerging. It may be due to the different meanings united under CE and numerous currently operating circular business situations. The CE area is an emerging field of research and has primarily concentrated on industrial products and circularity [6,32–36]. Trends in Sustainable Consumption (SC) are necessary to realize a sustainable society and economy [37]. SC meets market requirements, reducing the adverse effects of content extraction, processing, and usage [3,4]. Companies are potential enablers of SC in the CE by changing production processes and consumption patterns via addressing consumer needs in new ways [35,38]. In recent decades, several types of sustainability-focused companies have appeared. The business model's creativity is a systemic approach to company change [39,40]. They characterize business models as "a conceptualization of the way a company does business" to "identify the elements and relationships that describe a company's business." Circular business strategies have been summarized as "slowing, closing and narrowing" capital loops [38]. Slowing circles refer to product lifetime extensions and increased use and having direct ties with sustainable consumption. Lewandowski suggested a business model structure that integrates the circular economy concepts and includes PSS (Product Service System) as a circular business model [32]. PSS is proposed to lead to SC, possibly [41]. The circular business model literature emphasized resource conservation and business model innovation, implemented through reuse, repair, and remanufacture [42,43]. Circular business models are flexible and adaptable to the environment and capabilities of the enterprise [44], and multiple circular (and linear) business models may also operate at the same time [33]. If the operations upstream and downstream are 'circular,' a business model may be considered 'absolutely circular' [45]. The food industry has several harmful effects on culture and nature [46]. It accounts for roughly 30% of overall global energy consumption and about 22% of greenhouse gas emissions [47,48]. Every year, approximately 14% of the world's food is wasted before reaching the market, resulting in a USD $400 billion loss [49]. The United Nations has stated that food systems must be rethought, and inefficiencies such as food loss and waste must be addressed immediately [47,48]. Several of the UN's Sustainable Development Goals [50,51], such as Zero Hunger (Goal 2), Good Health and Well-Being (Goal 3), and Responsible Production and Consumption (Goal 12), are related to the food sector and have clear interconnections [52]. To ensure sustainable consumption and production patterns, is especially relevant to the food sector. By buying goods daily, consumers have a significant effect on the environment, but current and expected material usage rates are unsustainable. Growing consumption combined with expected middle-class growth in developing countries would necessitate even more capital [52–55]. Policy makers and academicians have made a considerable effort to tackle the population growth issue [56]. With the world's population expected to exceed 9.1 billion people by 2050 [57], the natural

resources needed to maintain current lifestyles will require the equivalent of nearly three planets. Food demand is expected to rise by 70% by 2050 [58,59], with consequences for food loss and waste.

The origin of goods is a significant factor in customer preference in the Italian market [30]. Consumers also demonstrate a greater willingness to pay for environmentally friendly consumption while providing information on the local origin or lower environmental impacts. Another essential knowledge drive factor is product certification. The availability of food certification information (organic food or product origin) will increase final demand [60–63]. The key motivation that affects customer preferences for natural food products is product health [64]. It has been seen primarily in the consumption of organic food. It was also noted that the purchase of environmentally sustainable products could be fostered by health concerns [8]. The combination of a health and carbon logo, for example, has a more beneficial impact than separate logos or no logo [65]. Sustainability is another compelling drive factor in shaping food opinions. Literature in recent years has focused on the "ethical consumer" who, through his purchasing choices, expresses a sense of responsibility towards society [66]. Research shows that a significant percentage of consumers are also prepared to pay premium prices for environmentally friendly products [67]. Consumers were particularly likely to purchase them when they were also labelled as having local origin regarding lower carbon footprint products [68]. In influencing SC, the brand is also highly significant. The brand differences play a role in determining products' attitudes with new ingredients regarding our analysis's focus [13]. The preference for organic food is another aspect that emerges in SC literature [69]. Literature states that the increase in demand for organic food is increasingly linked to consumer preferences for its lesser environmental effect, especially in Northern Europe [70]. Given a positive and growing trend in bio-food consumption, it would be essential to examine how customers think organic foods have a less environmental effect and people who purchase WTV goods hypothetically. Consumers are generally interested in reading nutrition data. In several studies, nutrition information changes products' value and increases the willingness to pay for improved outcomes [71,72]. Consumer concerns sustainability, labels, nutritional information influence respondents' choices [73]. Since the foodstuffs analyzed in this article will also have a higher nutrition content, it seemed appropriate to assess whether it might influence consumers' purchasing intention. One last factor deals with the broad evidence of gender's effect on the probability of buying environmentally friendly food. Women generally have a higher chance to buy organic food than men [74] since they are aware of and are more susceptible to food safety and health problems than men [75]. There has an engagement between the consumer of organic food products and the circular economy. However, the perception of the consumer differs based on their age. The highly qualified young people who lived in various Hungarian cities have the adequate attitude toward consuming organic food and are most conscious about the circular economy [76]. A certain group of people in Hungary follows the current trends and purchases organic food. The determinants of organic food consumption of the group are product freshness, healthiest diet, trust, etc. The positive health impact is the most dominant factor because it contains harmful ingredients [77]. Our research question is: how do the origin of products, information related to production/nutrition value, consumer education, and certain socio-demographic characteristics affect sustainable Waste-to-Value food consumption intentions?

**The hypotheses:**

**Hypothesis 1.** *Giving importance to the origin of the product and the product label (information on the production condition) positively impacts consumer purchase intention for sustainable WTV (waste-to-value) food products.*

**Hypothesis 2.** *The level of education positively influences purchase intention to the consumer of sustainable WTV (waste-to-value) food and mostly organic foods.*

**Hypothesis 3.** *There is a positive impact of respondents' socio-demographic characteristics (gender issues) on purchase intentions of sustainable WTV (waste-to-value) products.*

## 2. Materials and Methods

The study questionnaire consisting of closed-ended questions was developed by adopting items from relevant literature [78]. Based on the model specification, 9 items were included in the analysis. One dependent variable and 7 independent variables were included in the analysis. Has been added another item to check the validity. Items were expressed in numbers using the 5-point Likert scale, where 5 is an optimistic view (Strongly Agree), and 1 is an opposing view (Strongly Disagree) [79]. There are several questions with five Likert scale options such as (1) *Educated people should purchase WTV food product*; (2) *women are more health-conscious, and they want to buy healthy WTV product*. Some problems were scored in reverse to be consistent with the direction around the scale. The dependent variable for the logistic model was listed as a binary selection. In the survey period, the study measured the original variable on a 5-point Likert scale. The binary option (1-Yes) is measured by the highest values (4 and 5) of the Likert scale. The other binary options (0-No) are measured by the lowest values (1–3). This methodology is validated on practical and empirical grounds [78,80]. The respondents comprised of ages between 18 and 80 years, which were chosen based on convenience from the study area. Since the authors want to see consumer purchase intention of the sustainable WTV food products in the circular economy, the study area is restricted to certain parts of Europe. The authors chose the age range 18 to 80 years by different subgroups such as (18–26; 27–35; 36–44) to be more concerned about the current environmental condition and their judgment level. These usually reflect environmentally friendly products [81].

Further, a total of 544 questionnaires were collected at the end of the survey. Finally, the authors choose 499 cases for this study because of inconsistency in the rest of the questionnaire. Figure 1 provides the proposed model.

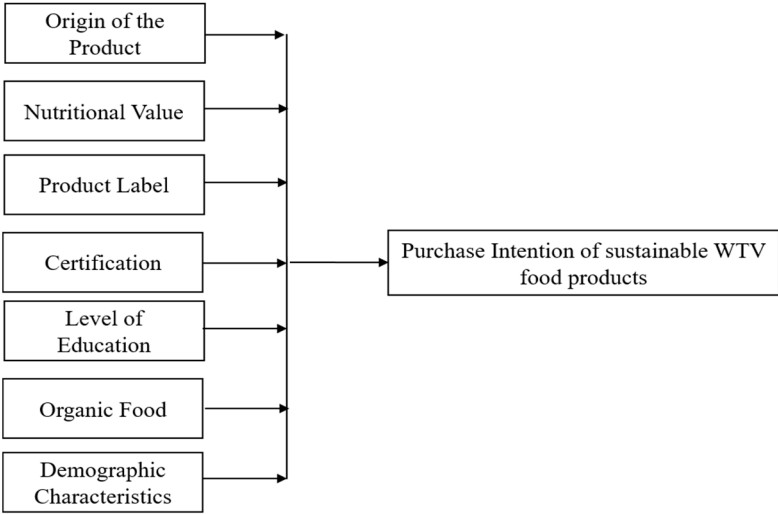

**Figure 1.** Proposed Theoretical Framework. Source: own edition.

*Research Design and Econometric Modelling*

Cronbach's alpha tests on all the variables were performed to assess their reliability as a combined scale [82]. The regression models were estimated to analyze factors affecting consumer purchase of sustainable WTV food products. To do this, the model uses the origin of the product, nutritional value consideration, product label, certification, brand, organic food, and socio-demographic characteristics of the population interviewed as independent variables. The empirical strategy that was adopted considered binary logistics regression model estimated in sequence with different model specifications to elicit factors affecting consumers' preferences for their choices. The model considers the probability of buying

a WTV food as a dependent variable if this could reduce production's environmental impact. The independent variable was calculated on a 5-point scale. Thus, each model's dependent variable was defined as a binary option by specifying the Likert scale's highest values (4 and 5) as 'yes' (1). The lowest values of the scale are (1 and 3) as 'no' (0). Even if dichotomization implies a potential loss of knowledge, this approach is justified on functional and empirical grounds [78,80]. Additionally, binary choice modelling promotes the analysis of the results. The general Equation (1) for the approximate conditional logic models is:

$$P_i(y_i \neq 0IX_i) = \frac{exp(X_i\beta)}{1 + exp(X_i\beta)} \tag{1}$$

where,

$i = 1, 2, 3, \ldots, n$;

$P_i$ = is the expected likelihood of a given option being made by person $i$.

$\beta_i$ = is an undefined parameter vector, and $X$ is a vector of explanatory variables representing the individual's characteristics and choices that are supposed to affect the respective option. Equation (2) shows the logistic regression model.

$$CPI = \alpha + \beta_1 OP + \beta_2 NV + \beta_3 PL + \beta_4 CoP + \beta_5 LoE + \beta_6 OF + \beta_7 DC \tag{2}$$

where,

CPI is the consumer purchase intention;

*NV* is the nutritional value;

*PL* is the label of the product;

*CoP* is the certification of the product;

*LoE* is the level of education;

*OF* is the organic food;

*DC* is the demographic characteristics.

Using Stata version 14, the theoretical structure was analyzed. First, the measurement model was used to assess the model's validity and reliability, and for the model fit and hypothesis, the later statistical model was evaluated.

We conducted depth interviews to clarify the questionnaire survey's details related to education and women's health-conscious consumer habits. We randomly selected 20 persons from previous respondents for the depth interview, regardless of age or gender. The interviews were conducted in a semi-structured individual system. The location of the interviews is Hungary, Budapest and Gödöllő. Date: January 2021. The same interviewer conducted the interviews.

## 3. Results

The measurement model provides the validity and reliability of the frameworks for quantitative measurements. Using Cronbach alpha [62], internal consistency was measured among the objects; the scale reliability coefficient ($\alpha$) is 0.84. The acceptable limit is '0.70' or higher. Table 1 shows the test result of validity and reliability.

**Table 1.** The test result of validity and reliability.

| | |
|---|---|
| **Average inter-item covariance:** | 0.0912041 |
| **The number of items on the scale:** | 9 |
| **The scale reliability coefficient ($\alpha$):** | 0.84 |

Table 2 shows the region of the respondents. All the respondents live in Hungary, but they moved here from different parts of the world. Among the 499 respondents, 263 belong to Europe, including Hungary.

**Table 2.** The frequency distribution of the respondents in the context of the Region.

| Region | Frequency | Percent |
|---|---|---|
| Asia | 88 | 17.64 |
| USA | 107 | 21.44 |
| Middle East | 25 | 5.01 |
| Europe | 263 | 52.71 |
| Africa | 16 | 3.21 |
| Total | 499 | 100 |

In the sample, the average age is 31 years. The highest frequency belongs to the age group 27–35 (Table 3). A total of 64% of respondents are female; this gender gap is considered quite realistic in the literature since women are more often responsible for grocery shopping instead men [80,83].

**Table 3.** The frequency distribution of the respondents in the context of the Age Group.

| Age Group | Number of the Respondent | Percent |
|---|---|---|
| 18–26 | 119 | 23.85 |
| 27–35 | 208 | 41.68 |
| 36–44 | 65 | 13.03 |
| 45–53 | 45 | 9.02 |
| 54–62 | 27 | 5.41 |
| 63–71 | 23 | 4.61 |
| 72–80 | 12 | 2.40 |
| Total | 499 | 100.00 |

Table 4 shows the correlation matrix. There is a clear positive connection between buying WTV food and the nutritional value that a customer considers before purchasing the product. There is also a clear positive connection between the intention and labelling of WTV food purchase, certification, organic food, and product origin. There is a weak negative association between gender and the purpose of buying food from WTV.

**Table 4.** Correlation Matrix.

|  | CPI | DC | LoE | PL | CoP | NV | OF | OP |
|---|---|---|---|---|---|---|---|---|
| **CPI** | 1 | | | | | | | |
| **DC (Gender)** | −0.35 | 1 | | | | | | |
| **LoE** | 0.05 | 0.00 | 1 | | | | | |
| **PL** | 0.81 | −0.30 | 0.05 | 1 | | | | |
| **CoP** | 0.88 | −0.29 | 0.08 | 0.76 | 1 | | | |
| **NV** | 0.89 | −0.32 | 0.07 | 0.74 | 0.84 | 1 | | |
| **OF** | 0.82 | −0.30 | 0.07 | 0.69 | 0.75 | 0.76 | 1 | |
| **OP** | 0.86 | −0.31 | 0.07 | 0.77 | 0.79 | 0.78 | 0.73 | 1 |

There was a negative relationship between the level of education and purchase intention for sustainable WTV food. The coefficient is statistically significant at a 5% level of significance. It is interesting in the case of graduate people, who do not favour adopting the new food products investigated, in line with the literature [12]. There has also a positive relationship between certification and purchase intention of sustainable WTV food products.

The Econometric model is

$$CPI = -7.76 + 3.68OP + 3.11NV + 1.97PL + 3.93CoP - 2.28Lo + 3.23\beta_6OF - 1.57\beta_7DC \tag{3}$$

Table 5 shows that all the coefficients are statistically significant at a 5% level of significance except demographic characteristics. The coefficients of the logistic regression show

the odds ratio. The coefficient of product label (*PL*) 1.97 indicates that a consumer buys the WTV food is 1.97 times higher for consumers who consider the product label than the consumers who do not believe the product label. According to the value of Pseudo-$R^2$, the model is the best predictor for data. The probability of Chi-square and Log-likelihood also confirms that the model is the best predictor for data. The value of $R^2$ is 0.9299 means that the dependent variable consumer purchase intention (CPI) 92.99% explained by independent variables (Equation (3)). This result should be read considering the questionnaire formulation, where some examples of certification are common in Hungary. It may have influenced replies that reveal a high propensity to Sustainable WTV products for consumers who are very attentive about specific certificates of product origins. There is a positive relationship between consideration of the nutritional value and consumer purchase intention of the sustainable WTV food products. They think that this could provide health benefits and a lower environmental impact, confirming the hypothesis made. Respondents reading food labels are more likely to purchase food from WTV. There was a strong positive relationship between the food label and purchase intention of the WTV food. It means that people who read product labelling have a higher possibility of buying this product. Giving high importance to certification when buying food has a positive effect on the effects of this probability. If they think this might have environmental benefits, respondents are more likely to purchase WTV food. Thus, it is exciting to appraise what factors could influence WTV products' purchase intention, motivated by environmental purposes. There was also a positive relationship between the origin and consumer purchase intention to sustainable WTV food products. Meaning that respondents who pay close attention to product origins, for example, the product is a local or imported product when buying food, seem to be more likely to buy sustainable WTV products. There has a positive impact on organic food and consumer purchase intention to sustainable WTV food products. That means respondents who think that organic food purchasing will reduce food consumption in the environment are more likely to purchase WTV items. Europe is the largest and most developed market globally for organic products, accounting for 54% of all global sales [84]. Thus, the eventual presence of a core of sustainable organic consumers interested in WTV food could represent target buyers for these productions. There was a negative effect of gender on consumer purchase intention to sustainable WTV food products. Women are generally more likely to buy organic food than men, as they are more aware of and sensitive to food safety and health issues than men [74,75].

**Table 5.** Logistic Regression Result.

| | | **Coefficient** | **Standard Error** | **Z-Statistics** | ***p*-Value** |
|---|---|---|---|---|---|
| | | **Independent Variable: Food Purchase Intention** | | | |
| | LoE | −2.28 | 1.05 | −2.18 | 0.03 ** |
| | PL | 1.97 | 0.86 | 2.29 | 0.02 ** |
| | CoP | 3.93 | 1.07 | 3.67 | 0.00 * |
| | NV | 3.11 | 0.92 | 3.39 | 0.00 * |
| | OF | 3.23 | 0.92 | 3.51 | 0.00 * |
| **Dependent** | OP | 3.68 | 0.94 | 3.92 | 0.00 * |
| **Variables** | DC (Gender) | −1.57 | 0.85 | −1.85 | 0.07 |
| | Constant | −7.76 | 1.48 | −5.26 | 0.00 |
| | Observation | 499 | | | |
| | Pseudo-R2 | 0.9299 | | | |
| | Probability > chi | 0.0000 | | | |
| | Log-likelihood | −24.07 | | | |

Note: * and ** indicate the coefficients are statistically significant at 1% and 5% level of significance respectively.

*Depth Interview and Feedbacks*

In the survey, we found interesting correlations between the gender issue and qualifications [73,74], which we wanted to explain by conducting depth interviews. The following responses were obtained on the twenty samples examined. For education-related questions (1), we hypothesized that educated consumers were more likely to buy healthier, waste-to-value foods, but responses to depth interviews confirmed the opposite. We found a similar opinion in 12 of the twenty respondents. Of these two interview questions, we present two typical answers each.

(1) Educated people should purchase WTV food products to promote a circular economy. Do you agree or disagree? Please explain your answer!

> *"No, I disagree with that. To promote the circular economy, we need everyone's participation in buying WTV products. It does not matter at all whether they are educated or not. Everyone should purchase WTV products."-student from India.*

> *"Disagree. It is not a matter of education; it is a matter of awareness. I think it has become a superstition that educated people always do the right thing. There is a huge difference between theoretical knowledge and its application in real life and its adherence. Suppose the government or other welfare agencies can induce a strong awareness about the benefits of purchasing WTV food and can make it a good habit for everyone. In that case, it does not affect much whether someone is educated or not."-student from China.*

In the case of young people, this knowledge is already acquired in primary school. Selective collection and avoiding food waste are not a matter of higher education. This fact was revealed during depth interviews. Male and female roles are much more homogeneous for those under 35 years of age than for other age groups. The function of motherhood does not appear as consciously as we think in terms of social norms. Respondents provided surprising answers to the question about women's health awareness during depth interviews (2).

(2) Women are more health-conscious, and they want to buy healthy WTV products. Do you agree or disagree? Please explain your answer!

> *"Since women go through unusual processes, for example, giving birth to a baby, unlike men, they are indeed health conscious. If they do not have sufficient money to buy WTV products, then I would answer a big NO. If the products become more affordable, they will buy such products."*

> *"I disagree, most of the women are not conscious because most of them do not consider WTV as a healthy product, due to the lack of awareness concerning this WTV food product. Therefore, there is a need to give information to women on the importance of WTV products for their health and in the circular economy."*

These answers could be the solution for our dilemma related to gender and education issues. In the case of single people, it does not matter what gender the consumer is. In the case of young people, women and men also play an equal role. Additionally, in the case of education, the important thing is that a healthy lifestyle is so commonplace for young people that it is no longer considered extra knowledge. Everyone knows who has a basic education. It is also important that everyone knows living healthy is cheaper than eating unhealthy things and spending a lot of money on healthcare.

## 4. Discussion

According to the literature, women are much more likely to buy healthy food, and education plays a very important role in helping someone choose a healthy lifestyle [78]. These characteristics are true for society, so for all ages. Food labels are also very important to consumers, and in connection with the consumption of local products, this is one of the most important characteristics, based on which the consumption of domestic, traditional foods is preferred [85]. According to the literature, female consumers also dominate organic

foods because they know healthy foods much better, are aware of their content values and make conscious purchases. The origin of goods is a significant factor in purchasing preference, with consumers being more willing to pay for environmentally friendly consumption while receiving information about local origins or lower environmental impacts. This local system is also strongly supported by the product certification system. Additionally, circular economic systems can only be sustainable structures with local market solutions [36,43,44].

Based on the survey conducted, we found that the importance of the origin of the product and the product label positively influences the consumer intention of sustainable WTV (waste to value) food products. According to our second hypothesis, the level of education has a positive effect on the consumer intention of sustainable WTV foods and especially organic foods. It was not confirmed by the questionnaire. Based on previous studies, the hypothesis should have been confirmed by the results of the questionnaire survey. We continued to explore the reason for the different survey results with a depth interview. Depth interviews revealed that young and highly educated consumers link knowledge related to waste management to basic education [73]. For this reason, education is not considered decisive in the preference for sustainable WTV foods. This conclusion requires further investigation because almost all of the respondents had tertiary education, so this aspect was not given sufficient weight among the criteria systems. So, respondents assume that the average consumer has a degree, which is certainly not true today. The third hypothesis is that the socio-demographic characteristics of the respondents have a positive effect on the purchase of sustainable WTV (waste to value) products. The research results did not confirm the hypothesis. In the case of the surveyed consumers, the younger age group (18–35 years) shows a greater preference for buying sustainable products. It is also a surprising result that gender characteristics in this age group do not influence consumption patterns. Women and men showed the same preferences. It is important to note that depth interview studies conducted to examine further the questionnaire research results also showed the same result. What was important behind the phenomenon was the fact that a significant proportion of respondents were or were living a 'single lifestyle' at the time of the study. In the case of the 'single lifestyle,' there was no difference in the preference for sustainable consumption by gender [74]. Additionally, important fact is that food-related jobs were feminized jobs, and that is changing today, so the role of women in food procurement is also changing [75].

## 5. Conclusions

Based on the questionnaire survey and depth interviews, it can be clearly concluded that the younger generation (18–35 years old) thinks fundamentally differently about sustainability than it has emerged from previous surveys. From the responses, it is clear that circular economic systems are linked to sustainability, less pollution and environmental risk. They acquired this knowledge related to the protection of the environment and sustainability in basic education. Therefore, they do not call higher education the criterion of sustainable consumer behavior. It is also very interesting that in the case of gender issues, there is no definite difference in the purchasing decisions of young women and men; basically, the same environmental awareness is characteristic of both groups. In terms of consumer gender, it does not follow the traditional value system, which is a favorable trend in terms of support for circular systems. The study's motive is to determine the consumer purchase intention towards sustainable WTV food products in the circular economy. A structured closed-ended questionnaire was administered to the consumers who moved to Hungary from different parts of the world. In this study, the authors analyzed the consumer purchase intentions toward sustainable WTV food products in the circular economy. Most consumers buy sustainable WTV food products willing to reduce the waste in the production process to reveal respondents' eventual preferences towards sustainable consumption. Regarding the willingness to buy WTV food, consumers who read food labels when buying and think that they could have environmental, or health benefits are more likely to state a positive purchase intention. Excitingly, a core of sustainable

consumers appears to be emerging who agree that purchasing organic food will reduce the environmental effect of food consumption and are more likely to buy sustainable food from WTV. Sustainable consumption and production of WTV foods are essential but not an adequate condition for a successful transition to a circular economy. Sustainability must become an overarching concept in all policies to switch from weak to strong, sustainable consumption policies. Researchers and their policies have been primarily concerned with how consumers can be influenced to establish sustainable food markets for WTV in the field of circular economy (CE). One lesson learned from this study based on the findings is that most consumers are aware of their nutritional value and purchase more organic food due to environmental and health benefits that can promote the circular economy.

This research is just an initial step towards evaluating future customer contribution to CE in the food sector. A prospective study should focus on a more extensive and nationwide representative sample to avoid the problems linked to self-selected and biased samples. Additionally, WTV products' production may help make this form of research more accurate, enabling customers to evaluate the effects to determine their sensory acceptance.

The nature of the research sample limits the drawing of conclusions. The survey revealed that respondents have a degree or are about to graduate. It is also a very important feature that the respondents are international consumers, and they are much more informed than the average consumer in terms of consumption habits. Basic literacy determines the responses that lead to careful conclusions. The article gave new results, similar to what has not been described in the literature so far, so it is difficult to justify the literature background. In the light of the above, the present research can be a starting point for further research programs that seek to clarify unusual contexts in relation to education and gender issues in relation to sustainable, value-creating consumption patterns. Care should be taken when generalizing the results because we may come to erroneous conclusions. Based on the study, we only formulated recommendations for further studies, which can be focused on analyzing the topic much more accurately.

**Author Contributions:** All authors: conceived the study, involved in its design, and participated in questionnaire design and field data collection. S.A. (Shahjahan Ali): prepared the data, performed the statistical analysis. C.F., S.A. (Shahjahan Ali), S.A. (Shahnaj Akter) drafted the manuscript together. C.F. read and commented on the first draft. All authors have read and agreed to the published version of the manuscript.

**Funding:** Special thanks to the Hungarian National Research, Development and Innovation Office –NKFIH (Program ID: OTKA 131925).

**Institutional Review Board Statement:** Not Applicable.

**Informed Consent Statement:** Not Applicable.

**Data Availability Statement:** Not Applicable.

**Conflicts of Interest:** The authors declare no conflict of interest.

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
