# Peer review of "Analysis of Circular Thinking in Consumer Purchase Intention to Buy Sustainable Waste-To-Value (WTV) Foods"

_sustainability, doi:10.3390/su13105390_

Round 1
Reviewer 1 Report
The theoretical framework is consistent and well structured. The references are abundant and cover a full range of journals, with a focus on those with impact factor.
The research is coherent. I would recommend to better highlight the objectives of the research and to justify the hypotheses based on the literature you have covered.
For the quantitative research, I would recommend to explain the correlations you have used. I see a table with the values obtained, but you have different scales (e.g. for the gender is a nominal one. What correlation did you use?).
For the qualitative research (the in-depth interview) I would recommend to present the findings in a more structured way. You can use frequency tables or even in the text you can stress the results of the entire sample ( e.g. 8/20 respondents said that...). When you present only two opinions and you conclude that the presumption is not supported is not enough.
Because you started with a theoretical frame presented as a model, and you formulated the hypotheses in causal manner (A influences B), you can duplicate it at the end putting on those arrows the values you determined in the econometric model. This is a suggestion, not a requirement.
Author Response
Dear Reviewer,
Thank you very much for the positive opinion! The suggestions are very good, which you shared with us. The results of the in-depth interview were structured, and the presentation of the answers was modified. The comments were very helpful, the introduction of the proposed changes significantly improved the quality of the paper.
The Authors
Reviewer 2 Report
- The survey needs to be displayed in the article, including how to code to assess the answer.
- Measurements regarding socio demographic also need to be clarified.
- The results of the research discussion are very inadequate to provide answers and a conclusion that the level of education has a very dominant impact
Author Response
Dear Reviewer,
Thank you very much for your suggestions. It was very helpful in improving the article. The description of the in-depth interview section was expanded and the circumstances of the response were explained. In the Discussion chapter, we made additions regarding the interpretation of the results. This study was not related to education, so we worked on this topic only tangentially. The results justify that, based on our research, we should start further studies in this area in the future. Conclusions that can be formulated based on the results of the present research have been rewritten in the text.
“Deep interviews revealed that young and highly educated consumers link knowledge related to waste management to basic education. For this reason, education is not considered decisive in the preference for sustainable WTV foods.” - This conclusion requires further investigation because almost all of the respondents had tertiary education, so this aspect was not given sufficient weight among the criteria systems. So respondents assume that the average consumer has a degree, which is certainly not true today.
The Authors
Reviewer 3 Report
The topic is interesting and relevant. Although there is plenty of material in terms of green consumption, relatively little has been studied from this angle. There are some issues with the manuscript that requires attention:
You have used multiple theories to advance your argument. There is a lot of discussion on the various theories to explain discrepancy between consumer thought and action. However, the transition from theory to operationalization of the theory in terms of construct/s is weak. It is important to provide more background about this construct. What are its features and how do these features affect purchase intention. Similarly, purchase intention has its components that need to be discussed and explored. Some of the discussions on these constructs was provided in the methodology. Rather, more should be described in the literature review section.
This point follows the previous one. Since much has been discussed at the theory level with less at the level of constructs, the discussion leading up to the hypotheses turns out to be weak. Specifically, some of the factors as indipendent variables: why so? Why is this variable a independent varialbe?
The model you have suggested is an over-simplistic one without the consideration of context. How will this model change when contexts change?
You mentioned qualitative and quantitative data when discussing your model. Please elaborate. How are both are realted to your model?
Overall, there needs to be tightening of literature review and better discussion of literature leading up to hypothesis development.
Authors need to add other implications. Right now there is only few implications. Current literature will help authors with additional implications. There may be policy implications especially with regard to marketing and promotion. Will "promoting the psychological thinkings that accompany green consumption behavior intention" really "increase the sustainable Waste-To-Value (WTV) foods value in commerce and national policy?"
Author Response
Dear Reviewer,
Thank you very much for the positive feedback. And we read your comments carefully. According to the processed literature, we followed the logic of the research, but the studies led to surprising results. This may be due to the fact that the examined target group has almost 100% higher education, and following their consumption habits is particularly important to follow a healthy lifestyle, because as a foreign consumer, their basic consumption habits are different from the culture in which we were interviewed. The willingness to follow a healthy lifestyle is much stronger because they want to stay healthy because of being abroad. In the previous survey, we analyzed the consumption habits of foreign university students (https://doi.org/10.3390/su11113052) and the impact of these consumption habits on the local consumption system (university restaurants, buffets). The research revealed that it was not the habits of consumers that changed in the first place, but the supply of restaurants changed according to the needs of consumers (students). It was also observed that the appearance of the foreign population caused a significant change in the case of local consumers as well. In this case, for example, consumption of larger quantities of vegetables or oat products became typical, or there was a significant increase in consumption of chicken instead of pork. The emergence of international consumption patterns can therefore result in quite surprising local effects that can change in any direction. Thank you very much once again for your helpful comments. The proposed changes have been made in each chapter. The parts that have been modified from the original content are marked in red. We also consider the modifications you propose to be justified in relation to the model, so at the end of the study we set limits on the use of the results and conclusions of the study.
The Authors
Reviewer 4 Report
General comment
Review of manuscript titled “Analysis of circular thinking in Consumer Purchase Intention to buy sustainable Waste-To-Value (WTV) foods”. The paper attempts to establish relationship between intention to purchase WTV and several consumer-related factors including demographics, attitude towards organic foods, origin of food, health, and nutrition etc. While the paper addresses a relevant topic that has the potential to contribute to the body of knowledge about consumers’ perception, and ultimately positioning of sustainable Waste-To-Value (WTV) foods, however, the paper has several limitations and needs to be re-written for clarity. The manuscript lacks focus as the background did not provide adequate context for the objectives and the methodology used. Hence, the results and the conclusions drawn from the study may not be valid. The novelty of the research relative to previous research, together with the contribution of this study to the advancement of research in the domain of Circularity was not indicated. Overall, the authors’ line of thought is difficult to follow as the grammar needs lots of improvement.
Abstract
Line 19-21: The primary objective indicated is not clear.
Line 23-25: The primary research question is also not clear. The statement: “Would the consumer buy a sustainable WTV food product that negatively affects the environment when it is produced?” is not clear. Of what value is the WTV food product if it negatively impacts the environment? Also, while this question was indicated as the primary research question in the abstract, this was not mentioned in the introduction. It is also not clear how this research question links to the study objectives.
Introduction
The introduction needs to be re-written for better clarity.
Line 51 to 52: shift in manageability of what?
Line 55 to 56: deposit of what? And high amount of what?
Line 61: “these epic food items” no reference was made to any food item in the preceding statement.
Line 62: reference to circular economy as a methodology needs to be addressed.
Information provided in line 82 to 86 belong to the methodology section.
No reference was made to the impact of educational attainment on consumers attitude toward food, yet, this hypothesis was tested in the study (hypothesis 2)
Hypothesis 1: More clarity on the product label. Does this refer to nutritional information?
Hypothesis 3: How will socio-demographic information positively impact purchase intent for WTV foods?
Materials and methods
This section lacks pertinent information.
Line 184 to 185: “By adopting items from relevant literatures” which literature are the authors referring to? No references were provided.
How was the dependent variable measured? On a 5-point Likert scale?
How many items were included in the questionnaire?
How was gender, education and other variables measured? How were these variables measured on a Likert scale?
Line 191-192: The age range is quite broad (18-80 years). In the preceding statement, the authors indicated using the convenience sampling approach, yet the authors mentioned that they chose respondents aged between 18 and 80 years due to their concern about current environmental condition and judgement level. This is contradictory.
The study involves human participants, it was not stated if ethical clearance was obtained for the study.
Statistical analysis
No proper justification was provided for the statistical analysis conducted.
Line 200 to 201: Why were the variables combined and the Reliability analysis (Cronbach’s alpha test) conducted? Was a dimensions reduction analysis (Principal Component Analysis) conducted on the questionnaire items before combining them together into one scale?
If the dependent variable (purchase intent) was measured on a Likert scale, an ordinal regression may be ideal instead of binary regression.
Results
It is not clear what scale was used to measure the independent variables, especially education level and gender. If these were measured on a nominal scale, the correlation analysis results presented is not valid, as this analysis is not appropriate.
The statement in Line 263 to 264 cannot be inferred from the results provided in the preceding lines.
Discussion
The results were not properly discussed against the backdrop of previous studies in this domain. Only one literature was cited in the discussion.
Conclusion
The conclusions drawn from this study are questionable considering the objectives, methodology, and statistical analysis lack detail and clarity.
Author Response
Dear Reviewer,
Thank you very much for your proposals. It was very helpful in improving the article. The suggestions are very supportive, which you shared with us.
The research question is included in both the abstract and the introductory part, which is as follows: „The first question is, “Would the consumer buy a sustainable Waste-to-Value (WTV) food product that negatively affects the environment when it is produced? That is, it does not come from a circular system?”
The research question is whether or not the consumer takes into account in his purchasing decision whether the food he purchased was produced sustainably. In the questionnaire, respondents were given a description related to WTV products, so they did not have to basically know this information. The primary goal of the research is to examine the consumption of WTV products and the knowledge of circular systems (CE) from a consumer perspective. The responses confirmed this well, but the results raise a much more exciting new area of ​​research on qualifications and gender issues. Indeed, the novel results led to conclusions that are not properly substantiated based on the literature. In this respect, we agree with the criticisms made. In order to better or more clear interpretation of the conclusions and results of the literature analysis, we have formulated limitations regarding the use of the results and conclusions.
The comments described have been taken one after the other and corrected in the text. Thank you very much for the detailed description of each problem. In each chapter, the details that have been corrected are marked in red. The corrections made on the basis of the proposals have significantly improved the quality of the article. Once again, thank you very much for taking the time to review the paper.
The Authors
Reviewer 5 Report
The paper covers a challenging topic of certain interest for researchers and academics.
The authors mentioned that the location of the interviews is Hungary, Budapest and GödöllÅ‘, while the date is January 2021. In order to increase the quality of the research, the authors must argue why they chose this particular country, implicitly these 2 cities from Hungary.
The authors should consider the following recommendations in order to improve the original manuscript:
- To include the structureof the paper in the Introduction section.
- It is more than necessary to include a new section "Literature review". The authors also did not provide sufficient evidence on literature review to support the hypotheses. The Introduction section also includes the Literature review section which is practically non-existent being mentioned only a few bibliographic references quite uncorrelated. Authors should take into consideration much more recent publications in the sphere of discussed subject matter, especially studies conducted during the last 5 years.
- Deepen the description of the limitations of conducted research and indicate the trends for further empirical research.
- To expand the managerial implications in the article.
- The conclusions section needs to be greatly improved and expanded.
- Human proofreading, English grammar and spelling correction are also required in order to improve the quality of the manuscript.
- I would also like to see a well-developed discussion comparing and contrasting solution/results presented in the work with existing work and then a subsection of it presenting contributions to theory/knowledge/literature and followed by a subsection on “Implications for practice”.
- The idea of sustainability needs to be more visible in this research paper (discuss the implication of the three main pillars: economic, environmental, and social).
- The sources must be added under each table and figure.
- The structured questionnaire must be included in the paper as Appendix, as the main tool underlying this empirical study.
- References in the text must be included following the instructions for authors based on the standards of Sustainability journal.
Regarding consumer behaviour framework, but also the most recent impact of Covid 19 pandemic on economy (which authors not even mentioned), I suggest extending the literature section by including at least the following relevant studies:
- Hawaldar, I.T.; Ullal, M.S.; Birau, F.R.; Spulbar, C.M. Trapping Fake Discounts as Drivers of Real Revenues and Their Impact on Consumer’s Behavior in India: A Case Study. Sustainability 2019.
- Timpanaro, G.; Bellia, C.; Foti, V.T.; Scuderi, A. Consumer Behaviour of Purchasing Biofortified Food Products. Sustainability 2020, 12, 6297.
- Aldieri L, Vinci CP. Green Economy and Sustainable Development: The Economic Impact of Innovation on Employment. Sustainability. 2018; 10(10):3541.
- Maryam Batool , Huma Ghulam , Muhammad Azmat Hayat , Muhammad Zahid Naeem , Abdullah Ejaz , Zulfiqar Ali Imran , Cristi Spulbar , Ramona Birau & Tiberiu Horațiu Gorun (2020): How COVID-19 has shaken the sharing economy? An analysis using Google trends data, Economic Research-Ekonomska Istraživanja, DOI: 10.1080/1331677X.2020.1863830 To link to this article: https://doi.org/10.1080/1331677X.2020.1863830
- Jing, X.; Guanxin, Y.; Panqian, D. Quality Decision-Making Behavior of Bodies Participating in the Agri-Foods E-Supply Chain. Sustainability 2020, 12, 1874.
I consider this article needs to be significantly improved because at this moment it does not meet the academic standards for publication.
Author Response
Dear Reviewer,
Thank you very much for the helpful comments you shared with us in the review. The authors chose these two cities for the survey because they are currently working here and it was easier to organize and control the research. In particular, conducting in-depth interviews. In the Introductory Chapter, we present the literature background, which focuses neither on the details of sustainability nor on the theoretical context of the circular economy. The aim is to present the details related to food procurement, which are detailed through 62 references. Many thanks to the reviewer for the suggested scientific references, three of which have been incorporated into the paper. The sources used are Q1 and Q2 publications, which have been supplemented with additional recent publications.
We made significant additions to the introduction, the methodology chapter, the result chapter, and the conclusions chapter based on the reviews. Changes and new sections are highlighted in red in the text. The grammar was run on the Grammarly program, and then a native-speaking colleague checked the final version. The tables were marked according to the guideline provided on the “Sustainability Journal” page, the References were created in Zottero and set to the MDPI format.
Comparing the opinions of the reviewers, we have improved the original version, thank you for both the praise sentences and the criticisms. We think the paper has improved a lot after the corrections have been made. The comments described have been taken one after the other and corrected in the text. Thank you very much for the detailed description of each problem. In each chapter, the details that have been corrected are marked in red. The corrections made on the basis of the proposals have significantly improved the quality of the article. Once again, thank you very much for taking the time to review the paper.
The Authors
Round 2
Reviewer 3 Report
The authors attempt to examine specific food consumption(e.g., Waste-to-Value (WTV) products) influences responses to purchase intention. I think this is a fascinating idea to research and commend the authors on an extensive literature review that was interesting and fun to read. I also liked the stimuli and general idea behind the two studies however there are still notable flaws within each of them.
In this study, the model of this research is not convincingly teased apart from a consumer purchase intention. In most purchase intention research, sustainability is compared against many different conditions. However, this study discusses the purchase intention of sustainability on an overly narrow topic. I believe authors have to consider the theoretical background of sustainable consumption.
Last, the hypotheses should be reworded and reorganized based on the literature review. As mentioned earlier, I don’t agree with theorizing of H1, H2, and H3. Authors should be justified the hypotheses with consumer research papers.
Selection of variables. The rationale of the selection of variables is missing in the paper. Specifically, why did the authors examine these factors? Will other possible factors lead to the same finding(e.g., income, desire for sustainability and customer experience so on)? It is also odd to introduce gender issues as the notable result in this paper. Overall, it is unclear how the variables examined in the paper are weaved together to form a coherent story. The authors should provide a strong rationale upfront for all the variables.
Author Response
Dear Reviewer 3,
Thank you very much for your valuable feedback. We examined the suggestions of the review step by step. We agree that the literature background needs to be better specified to formulate hypotheses. Therefore, the literature section was expanded and the hypotheses were reworded based on recent findings. These are marked in red in the text.
In the methodological part, we also made significant changes in order to justify the selection of variables. Logical causal relationships were written for point-by-point follow-up of interrelated factors. These parts are marked in red in the text.
Thank you very much for your feedback!
Best wishes,
The Authors
Reviewer 5 Report
The original manuscript has been significantly improved. The authors followed the recommendations included in the previous review report so that the quality of their research article has greatly increased. Despite its inherent limitations I already mentioned in the previous report, this research article is quite interesting and can be of great interest especially to researchers, PhD students, academics, the university environment. The revised version of the manuscript complies with the Sustainability journal standards. I also appreciate the hard effort of the authors in this regards.
Author Response
Dear Reviewer,
Thank you very much for your comments and words of praise. The article has been corrected and refined in some details. The literature section has been expanded and the limitations have been modified in part.
Thanks for the review!
Best wishes,
the Authors